# Waveguide-coupled nanopillar metal-cavity light-emitting diodes on silicon

V. Dolores-Calzadilla[1,†], B. Romeira[2], F. Pagliano[2], S. Birindelli[2], A. Higuera-Rodriguez[1], P.J. van Veldhoven[3], M.K. Smit[1], A. Fiore[2] & D. Heiss[1,†]

Nanoscale light sources using metal cavities have been proposed to enable high integration density, efficient operation at low energy per bit and ultra-fast modulation, which would make them attractive for future low-power optical interconnects. For this application, such devices are required to be efficient, waveguide-coupled and integrated on a silicon substrate. We demonstrate a metal-cavity light-emitting diode coupled to a waveguide on silicon. The cavity consists of a metal-coated III–V semiconductor nanopillar which funnels a large fraction of spontaneous emission into the fundamental mode of an InP waveguide bonded to a silicon wafer showing full compatibility with membrane-on-Si photonic integration platforms. The device was characterized through a grating coupler and shows on-chip external quantum efficiency in the $10^{-4}$-$10^{-2}$ range at tens of microamp current injection levels, which greatly exceeds the performance of any waveguide-coupled nanoscale light source integrated on silicon in this current range. Furthermore, direct modulation experiments reveal sub-nanosecond electro-optical response with the potential for multi gigabit per second modulation speeds.

[1] Photonic Integration, Department of Electrical Engineering, Eindhoven University of Technology, Postbus 513, 5600 MB Eindhoven, The Netherlands. [2] Photonics and Semiconductor Nanophysics, Department of Applied Physics, Eindhoven University of Technology, Postbus 513, 5600 MB Eindhoven, The Netherlands. [3] NanoLab@TU/e, Eindhoven University of Technology, Postbus 513, 5600 MB Eindhoven, The Netherlands. † Present addresses: Fraunhofer Heinrich-Hertz Institute, Einsteinufer 37, 10587 Berlin, Germany (V.D.-C.); Infineon Technologies, 93049 Regensburg, Germany (D.H.). Correspondence and requests for materials should be addressed to V.D.-C. (email: victor.calzadilla@hhi.fraunhofer.de) or to B.R. (email: b.m.romeira@tue.nl).

The development of high-density optical interconnects with reduced energy consumption has been identified as one of the major challenges in future computing and communication systems[1]. A new generation of photonic devices, integrated within or on top of a complementary metal-oxide-semiconductor (CMOS) chip, is therefore needed, featuring unprecedented levels of integration density, speed and energy efficiency[2,3]. For the light sources, the most promising performance on silicon has been achieved by ring lasers[4,5], based on III–V active layers bonded on silicon. These hybrid III–V/Si devices have however a relatively large footprint (several tens of μm$^2$) and present a power consumption far exceeding the requirements of future interconnects[1]. On the ther hand, in the last decade, a new class of nanophotonic light sources has emerged which use either photonic crystals or metallic layers to achieve strong optical confinement, eventually leading to ultralow-threshold lasing and other interesting effects, such as spontaneous emission enhancement via the Purcell factor[6]. While low-threshold photonic crystal lasers on InP have been demonstrated[7], their device footprint also remains relatively large (tens of μm$^2$) and integration on silicon has been demonstrated only very recently[8,9].

Metal-cavity nanolasers[10–12] can offer many advantages in terms of ultrahigh integration density, excellent cooling properties[13], and potentially ultra-fast modulation, but so far continuous wave operation of subwavelength lasers at room temperature has been achieved only with relatively high threshold currents ($\sim$mA) and unreported output power[14]. Additionally, waveguide coupling, modulation properties and integration on silicon have not been experimentally reported for metal-cavity lasers. Simple theoretical considerations show that aggressive scaling of metallic lasers well below the wavelength (therefore in the plasmonic regime) will result in unacceptably high threshold current densities[15]. In this context, the use of nanoscale light-emitting diodes (LEDs) instead of lasers for on-chip communication systems requiring low power consumption has been suggested[15], and indeed submicrometer LEDs have been demonstrated[16–19]. Unlike lasers, LEDs do not exhibit a threshold and can therefore be efficient at low current injection levels, do not require low-loss cavities and are therefore less sensitive to fabrication imperfections. Nano-LEDs can emit in a single spatial mode as nanolasers, and nanoscale LEDs with significant Purcell-enhanced spontaneous emission can in principle reach modulation bandwidths as high as lasers[20]. Furthermore, an LED-based nanophotonic source allows for a modulation speed beyond the 3 dB bandwidth without a substantial decrease in modulation depth as compared with lasers whose response deteriorates quickly beyond this point and large extinction ratios at high speeds can be maintained in nano-LEDs at low bias current levels, unlike a laser which requires high pumping conditions to reach large bandwidths. Despite the aforementioned advantages and progress in the field, the reported devices are not practical for photonic integration because they lack a low-loss output waveguide, mostly exhibit low output power below the nW level[18] and suffer low external quantum efficiencies in the $10^{-7}$ range[19], as reported for a photonic crystal LED and a plasmonic LED, respectively. For their application in photonic integrated circuits, such sources must be efficiently coupled to waveguide-based components. In this respect, a few waveguide-coupling schemes have been theoretically proposed[21–24], however there has been no experimental demonstration of practical implementations with the exception of Huang et al.[19], which used a lossy plasmonic waveguide.

In this work, we present a metal-cavity nanopillar LED on a silicon substrate working at telecommunications wavelengths (1.55 μm), coupled to an InP-membrane waveguide. With this approach we demonstrate full compatibility with the InP membranes on silicon (IMOS) integration platform[25], in which a III–V photonic layer provides the optical functionality and the silicon substrate hosts electronics. The devices show nW (tens of nW) measured output powers at $\sim$100 μA ($\sim$10 μA) current injection levels, at room and low temperature, respectively. When corrected for the outcoupling and setup collection efficiency, this corresponds to waveguide-coupled powers of 22 nW (300 K) and 336 nW (9.5 K) and an on-chip external quantum efficiency (EQE), $\eta_{QE}$, ranging from $10^{-4}$ (300 K) to $10^{-2}$ (9.5 K). Dynamic characterization via time-correlated single-photon counting measurements reveals sub-nanosecond electro-optical response and we confirm that such fast modulation is possible due to a strong non-radiative recombination. The reported data together with numerical simulations show the potential of metal-cavity nanopillar LEDs for efficient low-power interconnects operating at Gb/s data rates.

## Results

**Device design**. We conceived the device in a III–V membrane on silicon (IMOS) approach[25], which has been shown to enable a variety of functionalities including lasers[26], metal grating couplers[27], polarization converters[28] and demultiplexers[29], but we note that a similar approach could be implemented for coupling a InGaAs nanopillar to a silicon photonic waveguide[5], as proposed theoretically[21]. The diode cavity consists of a semiconductor nanopillar with an InGaAs active region as shown in Fig. 1a. The pillar is covered with a SiO$_2$ layer and then encapsulated with a silver cladding to form a metallo-dielectric cavity which defines an optical mode around 1.55 μm and suppresses a large fraction of the remaining radiation modes around the InGaAs active region, thereby making the coupling to the mode more efficient. Figure 1c shows the distribution of the mode of interest, which is well confined in the active medium and has high coupling efficiency with the fundamental quasi-transverse-electric (TE) mode of the waveguide shown in Fig. 1e as discussed later.

The metal cladding makes ohmic contact on a highly $n$-doped InGaAs layer at the top of the nanopillar, whereas a lateral $p$-contact is deployed over a large area on $p$-doped InGaAsP next to the pillar to minimize its contact resistance. The dielectric layer has a two-fold purpose, it provides electrical isolation between the diode contacts and also reduces the metal loss by decreasing the penetration of the cavity modes into the silver. The cavity modes in the nanopillar are evanescently coupled to a 450 nm wide InP waveguide, which isconnected to a grating coupler for characterization purposes.

We performed three-dimensional finite-difference time-domain (FDTD) simulations to calculate the spontaneous emission coupled to the waveguide (see Methods). We first considered an ideal cavity with a 1.5 μm high nanopillar with horizontal cross section of $325 \times 325$ nm$^2$, vertical sidewalls, 175 nm thick SiO$_2$ cladding, and encapsulated in silver. Refractive index values for Ag were taken from Johnson and Christy[30]. The average power emitted by dipoles in the InGaAs region and coupled to the waveguide, normalised to the power emitted in an unstructured bulk medium, is plotted on a logarithmic scale as a function of wavelength in Fig. 1b (green curve). Clear emission resonances are present at 1,568 nm and 1,482 nm, corresponding to the mode of interest with large waveguide coupling efficiency, and a vertical resonance (that is, vertically resonating between the top of pillar and the bottom SiO$_2$ buffer layer), respectively. The profiles of the electric field squared of

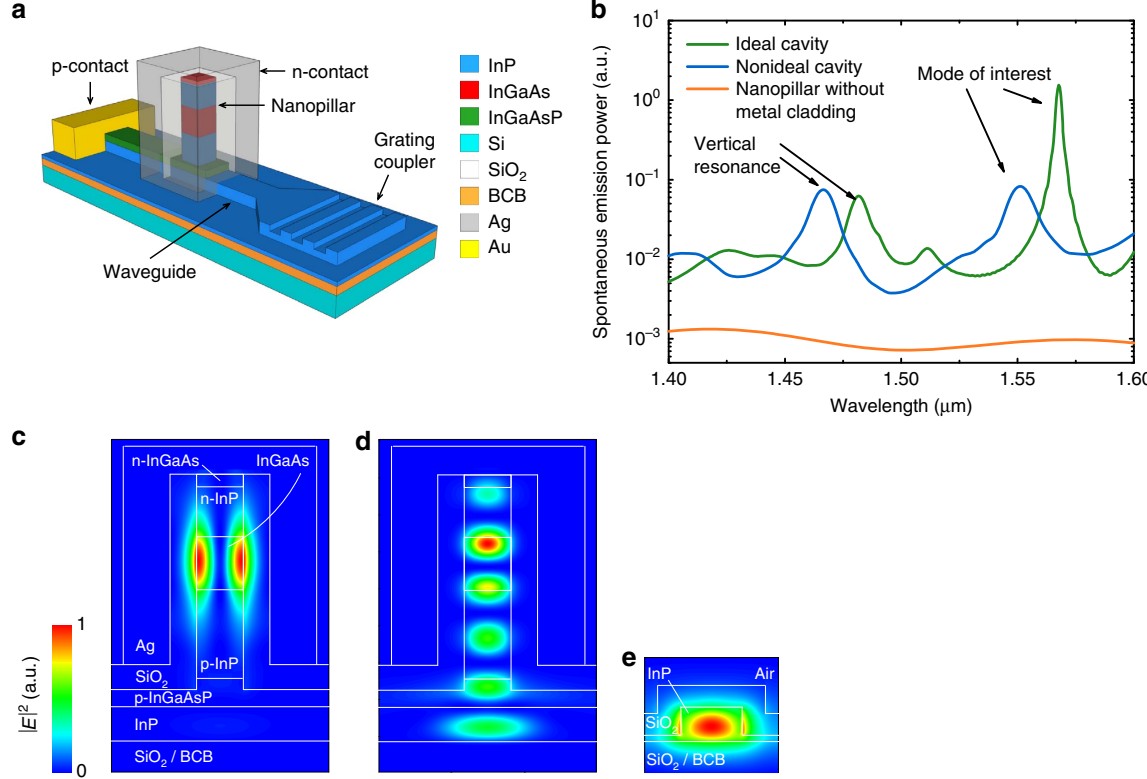

**Figure 1 | Design of the nanopillar LED device coupled to an InP-waveguide. (a)** Schematic representation of the nanopillar LED on a silicon substrate. The layer stack from top to bottom is: $n$-InGaAs(100 nm)/$n$-InP(350 nm)/InGaAs(350 nm)/$p$-InP(600 nm)/$p$-InGaAsP(200 nm)/InP(250 nm)/SiO$_2$/ BCB/SiO$_2$/Si. **(b)** Calculated spontaneous emission power coupled into the fundamental quasi-TE mode of the waveguide, as a function of the dipole wavelength. The plot shows the case of an ideal cavity (that is, low loss silver and vertical sidewalls), a nonideal cavity (that is, lossy silver and 2° sloped sidewalls due to etching), and a nanopillar without metal cladding and vertical sidewalls. **(c–e)** Normalized $|E|^2$ spatial distribution of resonant modes along the waveguide direction for the ideal cavity case, **(c)** and **(d)**, and waveguide fundamental mode **(e)**. **(c)** Resonance at 1,568 nm with dominant in-plane polarization and high waveguide coupling efficiency. **(d)** Vertical resonance at 1,482 nm. **(e)** Quasi-TE waveguide fundamental mode excited by the cavity resonance at 1,568 nm.

these cavity modes are shown in Fig. 1c,d. We note that the emission is strongly suppressed out of resonance, while the emission rate in the modes is slightly enhanced as compared with the value in the bulk. Indeed, we calculate a Purcell enhancement of 12 for the mode of interest at 1,568 nm in the ideal case of a monochromatic dipole, but we expect this factor to be reduced to a value close to one when taking into account the homogeneous and inhomogeneous broadening in the bulk active region[31]. The combination of moderate spontaneous emission enhancement into the mode and strong suppression of the emission into other modes, as enabled by the metallo-dielectric cavity, is the key approach to obtain efficient funneling of spontaneous emission into a guided mode. For comparison, the orange curve in Fig. 1b reports the power coupled into the waveguide in the case where no metal is present around the pillar, showing very poor coupling. By dividing the power emitted in the waveguide by the total power emitted by the dipoles (not shown), we estimate an average 'spontaneous emission coupling efficiency' (fraction of emitted photons which couple to the waveguide) of ∼15% around the resonance at $\lambda = 1,568$ nm. This corresponds to the efficiency of an ideal device where all recombination is radiative and the emitters are all spectrally resonant with the mode. We note that the emission rate into the mode strongly depends on the spatial and polarization matching between the dipole and the modal field. For dipoles well coupled to the mode the spontaneous emission coupling efficiency is calculated to be as high as 29%, corresponding mainly to the product of the optical efficiency (fraction of generated photons in the nanopillar mode that leave the cavity) and the waveguide coupling efficiency (fraction of photons leaving the cavity that couple to the waveguide mode), as the spontaneous emission into other modes is negligible at resonance. However, a part of the active volume is not well coupled to the mode, making the coupling of spontaneous emission to the waveguide less efficient on average.

We also considered the influence of nonidealities on the emission. Figure 1b shows the emission into the waveguide for a cavity with sloped sidewalls under a 2° as observed in the fabricated structures. Additionally, we employed another widely used Ag material model[32], which exhibits higher losses. The result is shown in Fig. 1b (blue line). Due to the sloped sidewalls, the mode overlap with the InP material increases which causes a blue shift of the peaks to 1,467 nm and 1,550 nm. The power emitted in the vertical resonance is not changed significantly, while the emission of the mode of interest around 1.55 μm drops which we attribute mainly to a reduction of the spontaneous emission rate into the mode due to the loss in quality factor. The corresponding average spontaneous emission coupling efficiency is reduced to about 7% for the mode of interest.

For collecting the output power, the device is connected to a grating coupler. Since the resonant wavelength is highly sensitive to the cavity size, a TE broadband grating coupler was designed to account for fabrication inaccuracies. This coupler performs the chip to free-space outcoupling with an efficiency

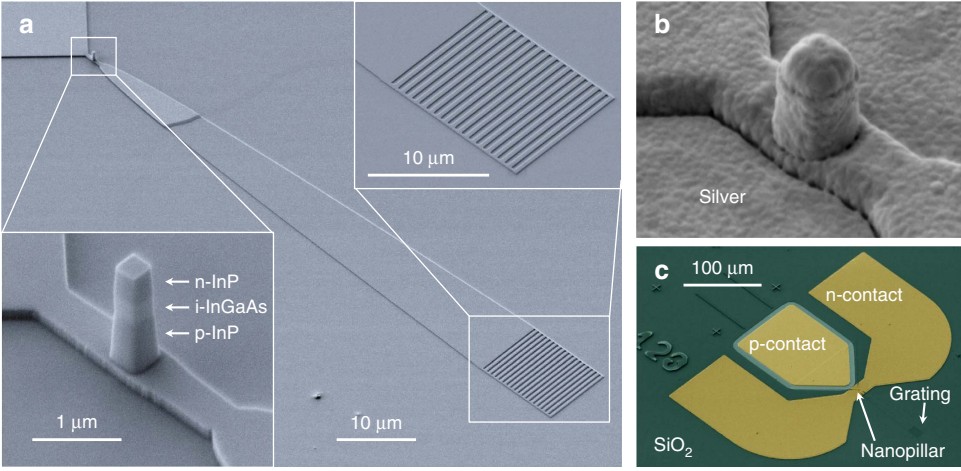

**Figure 2 | Scanning electron microscope images of fabricated devices.** (**a**) False-coloured image showing the fabricated device structure before metallization. The nanopillar lies on top of a waveguide connected to a grating coupler. (**b**) Metal-coated nanopillar after silver evaporation and rapid thermal annealing, before the sputtering of gold (100 nm) to prevent silver oxidation. (**c**) False-coloured image of the device after metallization, showing the electrical contacts.

higher than 40% and diffraction angles lower than 24° in the wavelength range 1.4–1.6 µm (Supplementary Note 1).

**Device fabrication.** The device was fabricated in a III–V layer stack bonded to a silicon substrate by means of an adhesive bonding technique using a low refractive index benzocyclobutene (BCB)[5]. At first, three electron-beam lithography steps are used to pattern and etch the nanopillar, followed by the waveguide and the grating coupler. Later, metallization steps for the electrical contacts are realised by optical lithography. As the cavity resonance frequency is very sensitive to nanopillar size, we fabricated a series of devices with slightly different pillar cross sections (the experimental results on the static and dynamic characterization were obtained on nanopillars with side length of 350 nm and 340 nm, respectively). Examples of fabricated devices are shown in Fig. 2 before and after metallization. For the $p$-contact we used an Au/Pt/Ti metallization scheme on $p$-doped InGaAsP ($P = 2.4 \times 10^{19}$ cm$^{-3}$), whereas for the $n$-contact we used an Au/Ag/Ge metallization on $n$-doped InGaAs ($N = 1 \times 10^{19}$ cm$^{-3}$) which we recently demonstrated to provide low optical-loss ohmic contact[33]. We measured contact resistances of $\sim 7 \times 10^{-4}\,\Omega\,\text{cm}^2$ and $\sim 5 \times 10^{-7}\,\Omega\,\text{cm}^2$ for the $p$- and $n$-contact, respectively. An extensive description of the fabrication process can be found in the Supplementary Note 2. The total footprint of the device is 280 µm × 170 µm which is mostly determined by the contacts footprint as can be seen in Fig. 2c. This design was required as determined by the spacing of the probes in the characterization setup. However, in a practical application the contacts can be made as small as the technology for electrical interconnecting to CMOS chips allows, which, together with the sub-micrometer footprint of the pillar, would enable very-large-scale integration levels.

**Static characterization.** We carried out the optical characterization in a micro-electroluminescence (EL) setup at low and room temperatures (see Methods). The devices were contacted electrically with RF probes, the light emission was collected from the grating coupler with a high numerical aperture (NA) microscope objective (50×, NA = 0.42) and measured by a calibrated powermeter or spectrally dispersed and detected with a cooled InGaAs array. The measured room-temperature

electrical characteristics of the diodes are shown in the inset of Fig. 3a. The diodes exhibited a constant reverse-bias current of 0.5 µA and showed a high series resistance of about 22 kΩ due to the high $p$-contact resistance since this contact was not annealed.

Figure 3a shows the spectrum at different bias points (indicated by arrows in the $I$–$V$ inset). We observed a carrier-induced blue shift of up to 0.1 nm µA$^{-1}$. For low pumping conditions, only the low-frequency cavity resonance at 1,557 nm is visible due to the better overlap with the luminescence spectrum of the bulk InGaAs, whereas an additional resonance shows up at 1,425 nm for higher pumping currents. The spectral position of the two peaks qualitatively compares well with the resonances in Fig. 1b (blue line). We note that the details of the measured spectra are not captured in the model of Fig. 1b since the latter does not consider the carrier distribution, the homogeneous broadening, and the change in refractive index due to current injection and local temperature.

Figure 3b shows the light-current ($L$–$I$) curves at different temperatures and the inset shows the corresponding on-chip EQE. The $L$–$I$ characteristics of the tested nanocavities (additional measurements are provided in the Supplementary Note 5) reveal that devices with identical horizontal cross sections display similar performances in terms of maximum optical output power and efficiency indicating a high yield. The maximum measured output power is nearly 4 nW at room-temperature and increases to 60 nW at 9.5 K. Accounting for the fact that we are collecting from only one side of the waveguide, for the grating out-coupling efficiency ($\sim 50\%$, see Supplementary Note 1) and for the measured setup collection efficiency (71%), this corresponds to waveguide-coupled powers of 22 nW (room temperature) and 336 nW (9.5 K). At cryogenic temperatures, a larger output power is obtained because the non-radiative recombination rate is considerably lower and therefore the radiative efficiency is higher. The experimental results are fitted adequately with a rate equations model which explicitly accounts for homogeneous broadening and for the imperfect matching between active material and cavity mode (Supplementary Note 7). Since thermal effects are not considered in the model, it does not fit the thermal roll-off at high injection levels.

The on-chip EQE (total number of photons coupled to the waveguide divided by number of injected electrons), shows

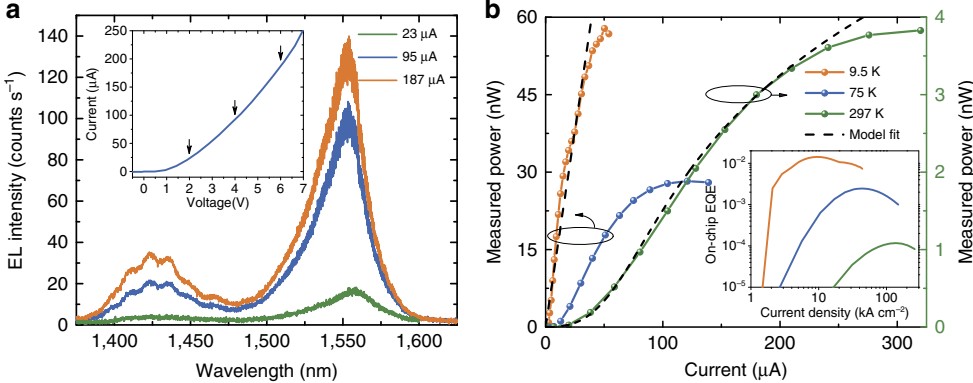

**Figure 3 | Static characteristics of nanopillar LED devices. (a)** Emission spectrum at room temperature for different bias currents. The inset shows the current–voltage characteristics and the arrows indicate the bias points used in the spectra. **(b)** Light–current characteristics of the metal-cavity LEDs at different temperatures (9.5 K and 75 K: left axis; 297 K: right axis) including a numerical model fit (dashed curves). The inset shows the calculated external quantum efficiency for the on-chip waveguide-coupled power.

a maximum which varies from $\sim 10^{-4}$ (at RT) to $\sim 10^{-2}$ (at 9.5 K). The low-temperature EQE is in the same order but smaller than the average spontaneous emission coupling efficiency of $\sim 0.07$ calculated with the realistic parameters used for the blue line in Fig. 1b. This is related to the fact that, even at low temperature, the spectral distribution of carriers in the bands does not perfectly match the mode spectrum, and presumably to a Ag loss higher than the one reported in Palik[32]. The latter effect is also confirmed by the fact that the experimental linewidth of the resonance at low temperature (25 nm) is larger than the calculated value of 14 nm.

The difference between room- and low-temperature efficiency is largely due to the effect of nonradiative recombination, dominant at RT as observed from time-resolved photoluminescence experiments (Supplementary Note 3). Specifically surface recombination plays a dominant role, as confirmed by the model fit of $L$–$I$s at both room and low-temperatures (dashed curves in Fig. 3b) using the surface recombination velocity as a fitting parameter (Supplementary Note 7). Additional contributions to the efficiency reduction may come from the broader thermal carrier distribution and from an increased attenuation coefficient in Ag. The change in Ag optical loss with temperature is confirmed by a larger linewidth of 42 nm observed at room temperature.

**High-speed dynamic characterization.** Time-resolved electroluminescence experiments were carried out to determine the modulation speed characteristics of our devices (see Methods and Supplementary Note 4). We directly modulated the nanopillar LEDs using a pulse pattern generator with a periodic pulse train of 80 MHz with pulse widths varying from 1 ns to 100 ps. Figure 4a shows the detected modulated optical output when the device was d.c. biased with a current of 11 µA and modulated with a small amplitude pulse.

The on–off switching time of the device is in the sub-nanosecond regime, and the switch-off can be fitted with a single exponential decay curve allowing to estimate a lifetime of $\tau = 289 \pm 3$ ps. The results in Fig. 4a can be compared with the slow exponential decay time ($\tau_b \approx 2.7$ ns) measured from the time-resolved photoluminescence of the InGaAs bulk material, in a test sample prepared with an identical epitaxial layer stack employed in the fabricated LEDs (Supplementary Note 3). The results indicate that minority carrier lifetimes in the metal-cavity nanopillar are much shorter than the nanosecond-range

lifetimes in the bulk material which otherwise would limit the nanopillar LED frequency response to a 3-dB modulation bandwidth $\sim 60$ MHz ($f_{3dB} \approx 1/2\pi\tau_b$). The lifetimes measured by time-resolved photoluminescence on nanopillars in the test sample of comparable dimensions to the nanopillar LED devices show recombination lifetimes varying from 324 to 184 ps for pillars with side lengths of 0.59 µm and 0.388 µm, respectively, which compare with the sub-nanosecond recombination lifetimes measured in Fig. 4a for a nanopillar LED. Furthermore, time-resolved electroluminescence measurements at low-temperature of the nano-LEDs (Supplementary Fig. 10) show a lifetime of $\sim 1.1$ ns, confirming that non-radiative recombination strongly increases with temperature. These results, together with the small calculated Purcell enhancement factor, confirm that the main physical process explaining the fast carrier dynamics measured in the nanopillar LEDs is non-radiative recombination due to surface states in the pillars with sub-micrometer cross section. The experimental electro-optical modulation results are also in good agreement with the numerical results (dashed curves in Fig. 4a) employing a rate equations model that takes into account both radiative and non-radiative recombination processes (Supplementary Note 7), further confirming that the sub-nanosecond response measured in the device is dominated by carrier dynamic effects.

Although the efficiency of the LEDs can be limited by such non-radiative process, this characteristic leads to the possibility of spontaneous emission light sources operating at modulation bandwidths beyond the limitations of the slow radiative recombination process of the semiconductor material. To show the potential of the nanopillar LEDs for modulation at GHz frequencies, we directly modulated the device using a pulse pattern generator in a Return-to-Zero modulation scheme with a periodic pulse train having a 50% duty cycle at repetition rates ranging from 2 to 5 GHz (corresponding to a pulse width of 250 ps and 100 ps, respectively). Figure 4b presents the optical output showing that the nanopillar LED output replicates the injected on–off periodic bit sequences. The bit stream has clearly resolvable off-pulses even at the higher frequencies of 5 GHz. At 2 GHz the optical response has a modulation depth of 96%, whereas it decreases to 42% at 5 GHz. This is a typical behaviour in directly modulated LEDs following a single-pole response of the form $H(f) = A_0/(1 + jf/f_{3dB})$ ($A_0$ is the electro-optical conversion efficiency), where the modulation depth decreases monotonically for higher modulation frequencies. While in the pumping range used in these experiments

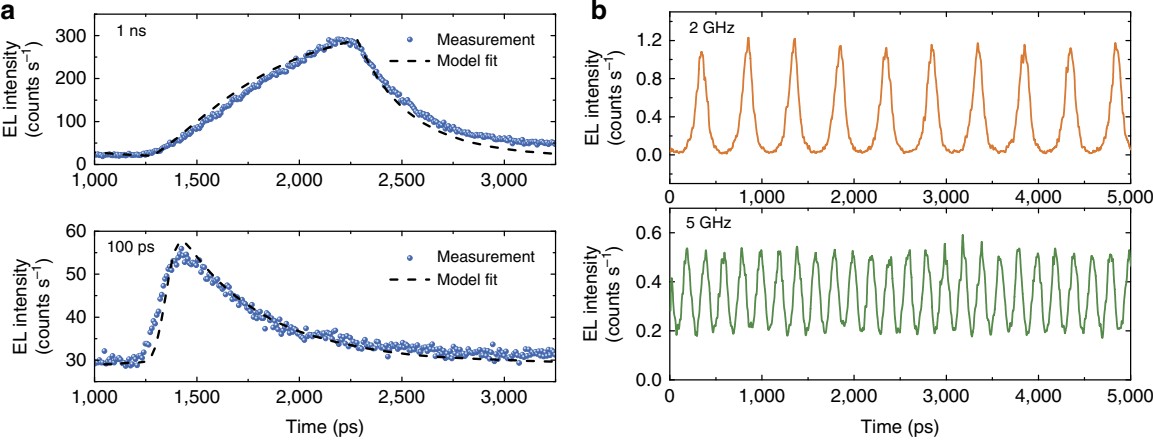

**Figure 4 | Dynamic characteristics of nanopillar LEDs. (a)** Time-resolved electroluminescence showing the electro-optical modulation response of the nanopillar LED to electrical pulses with widths of 1 ns and 100 ps at room temperature. (**b**) Direct modulation at room temperature with a pulsed pattern with 50% duty cycle at 2 and 5 GHz. The device was modulated with a peak-to-peak voltage signal of 1.4 V on a d.c. bias of 0.5 V.

the energy consumption of the device was in the range of 50–100 fJ per bit (Supplementary Note 6), as discussed in the next section, improvements in the design and fabrication indicate interesting prospects of achieving further reduction of the energy consumption close to the target values of optical interconnects (∼10 fJ per bit[1]) while improving the efficiency without substantial degradation of the maximum operation speeds reported here.

## Discussion

In summary, we present a waveguide-coupled nanopillar LED device with metal-cavity, fabricated in a III–V layer stack bonded to a silicon substrate. The device shows relatively high on-chip external quantum efficiency ($10^{-4}$ to $10^{-2}$ for room-temperature and 9.5 K, respectively) and output power (that is, nW to tens of nW) far exceeding previous nanoscale LEDs exhibiting up to hundreds of pW output[18]. Although some promising photonic crystal lasers have also been integrated on silicon substrates recently[8,9], metal nanocavities still offer a few advantages that justify a great interest and further investigation on this approach including reduced footprint and better heat dissipation[13], since the metal layer helps to spread and sink the heat in an efficient manner. We note that the power level achieved at low temperature (>50 nW) corresponds to over 400 photons per bit at 1 Gb s$^{-1}$, far above the shot-noise limited sensitivity of an ideal receiver[34]. With the expected low loss of short-distance interconnects and continuous progress in integrated receivers, this power level may enable intrachip data communications with an ultracompact source. Furthermore, the percentage-level efficiency at low temperature reported here suggests the potential of metal nanocavity light sources to perform very efficiently even at room-temperature if the non-radiative recombination processes (found to be dominant in our devices) are suppressed by better passivation techniques[35]. Very recently, our group developed a surface passivation method[36] based on wet chemical ammonium sulfide treatment and SiO$_2$ encapsulation that shows a reduction of the surface velocity by two orders of magnitude (∼500 cm s$^{-1}$) for nanopillars with similar dimensions and identical epitaxial layer stack to our nano-LEDs. As discussed in the Supplementary Note 7, this could provide up to 100-fold increase of the efficiency of our nano-LEDs reducing substantially the energy consumption. Additionally, the energy

consumption can be reduced by improving the ohmic contacts, using heavily doped InGaAsP layers[37] and a thicker p-doped InGaAsP layer.

Furthermore, an improved spatial matching of the active region with the optical mode may enable an increased coupling of spontaneous emission into the cavity mode and thereby increase the efficiency above the 1% level. This can be achieved without substantial modifications of the currently proposed metal-cavity nanopillar devices. Possible strategies include the reduction of the bottom InP post thickness, and the reduction of the insulator thickness, both resulting in increased mode confinement in the active region. A higher verticality (sidewall angle <1°), as reported in the literature using Cl$_2$:H$_2$:Ar inductively coupled plasma reactive ion etching[38], would also increase the quality factor.

We explored also the dynamic characteristics of the devices and found sub-nanosecond lifetimes, related to surface recombination at the pillar sidewalls. While nonradiative recombination affects the radiative efficiency, it is instrumental for achieving fast on–off switching, as previously suggested for the case of Auger quenching[39]. We demonstrated this by modulating the nanopillar LEDs with a pulse pattern generator at frequencies up to 5 GHz. In these results the nano-LED was operated at low current bias pumping levels (<40 μA) to avoid thermal heating and possible failure. Under these conditions, the surface velocity is the dominant non-radiative mechanism, however, as discussed in the simulation results presented in Supplementary Note 7, the Auger mechanism starts to dominate at moderate and high bias pumping conditions and could also be exploited in our nano-LEDs to achieve GHz operation speed.

Besides the methods based on fast non-radiative processes such as surface velocity or Auger quenching[39], other methods including tailoring the spontaneous emission by modifying the local density of optical states using 2D materials[40] or by leveraging the phase-change of a vanadium dioxide nanolayer[41], or using Purcell enhancement of the spontaneous emission can be exploited. In future designs an optimized emitter-cavity spatial matching combined with a mode volume reduction may result in significant Purcell enhancement of our nano-LEDs, enabling high modulation speed without compromising the radiative efficiency. Analysis of a Purcell-enhanced metal cavity nano-LED with optimized structure and assuming physical dimensions smaller than our current devices, and with improved surface passivation (Supplementary Note 7), reveals the possibility of operating

nano-LEDs at $\sim 1$ Gb s$^{-1}$ rate with on-chip optical power levels above 1 µW (corresponding to $\sim 10^4$ photons per bit) and energy consumptions $< 20$ fJ per bit. The results clearly suggest that further downscaling of our nano-LEDs could provide substantial improvements in terms of bandwidth and energy consumption per bit.

At last, it is possible to explore other methods that do not require Purcell enhancement to further improve the speed of the nano-LEDs, for example, taking advantage of Auger recombination at high carrier densities or using reverse-biasing of the nano-LED during the turn-off cycle to shorten the minority carrier storage time[42]. For example, we achieved a fast switch-off of around 123 ps using reverse-biasing of our nano-LEDs at room temperature (Supplementary Fig. 11), a value beyond the limitation of the non-radiative recombination rates reported in Fig. 4a.

These results are encouraging for future high-density optical interconnect systems requiring Gbps data rates at ultra-low power consumption, which could be achieved with arrays of directly modulated integrated nanoscale sources[2].

## Methods

**Simulations.** We used the FDTD method (three dimensional simulations) for the optical design of the metal-cavity nanopillars LEDs and the grating couplers. To calculate the spontaneous emission power coupled into the waveguide, we performed a series of 135 simulations, where we vary the position and orientation $(x,y,z)$ of a single electric dipole emitting in the range 1.4–1.6 µm within the active medium InGaAs, and monitor the power coupled to the fundamental TE mode of the waveguide. As the cavity is symmetric along and perpendicular to the waveguide (that is, in $x$ and $y$ directions), we only place dipoles in one quarter of the structure on a grid of 80 nm at 3 positions in $x,y$ direction and 5 positions in $z$ direction. The calculated waveguide-coupled emission was averaged over all positions and dipole orientations. The spontaneous emission coupling efficiency was calculated by normalizing the power coupled into the waveguide fundamental mode to the total power emitted by the dipoles placed inside the nanopillar cavity, while the Purcell enhancement factor was estimated by dividing the total power emitted by a dipole to the emitted power of a dipole in homogenous InGaAs[43].

The design of the grating coupler was performed with two-dimensional FDTD simulations. Since the waveguide width is much larger than the light wavelength at the grating coupler region, 2D simulations are sufficient to correctly describe the chip-to-free space coupling. We considered a grating pitch of 680 nm with 50% filling factor and 100 nm etch depth, on a 250 nm thick InP waveguide layer and SiO$_2$(50 nm)/BCB (1800 nm)/Si as substrate. The quasi-TE fundamental mode of the waveguide was used as optical excitation and the off-chip diffracted electromagnetic field is calculated at the top of the grating coupler. The out-coupling efficiency of the grating is calculated by measuring the power diffracted upwards. Later, the electric field is Fourier-transformed to get the angular distribution and the diffraction angle.

**Time-resolved electroluminescence setup.** The ultrafast dynamic characteristics of the metal-cavity nanopillar LEDs were measured using a confocal micro-electroluminescence (µEL) setup together with a time-correlated single-photon counting module for time-resolved measurements (Supplementary Note 4). A signal pulse generator (Agilent 81134 A or Anritsu MP1701A) was used to generate a short electrical pulse to excite the devices. A high-frequency bias-T (0.2 MHz–12 GHz) was employed to provide both an a.c. and a d.c. drive signal to the devices, contacted by 40 GHz high-speed probes with ground-signal-ground configuration (100 µm pitch).

The light emitted from the waveguide-coupled LED device was collected in free-space from the device's integrated grating coupler using a high numerical aperture (NA) objective (50×, NA = 0.42). The image on the sample plane was projected onto a single mode optical fiber (SMF-28J). The fiber collects the light from a limited circular area ($\sim 1.7$ µm diameter) of the grating coupler and guides it either to a Horiba FHR 1000 monochromator equipped with an InGaAs cooled detector camera or to a single photon counting detector. For measuring the light-current characteristics, however, the emission from the full grating coupler was measured with a large area photodetector placed after the microscope objective.

A piezo-electric scan stage was used to scan the sample. Both sample holder, piezo stage and electrical probes were placed inside a liquid helium flowing cryostat chamber allowing electro-optical measurements down to $\sim 9$ K. By using the InGaAs detector it was possible to acquire the luminescence spectra, while a Scontel superconducting single photon detector (SSPD) housed inside a cryogenic dipstick at 1.5 K and working at a bias current $I_b \sim -19$ µA with quantum efficiency of $\sim 20\%$, was used to measure the temporal decay. Before detection,

a spectral filter (FWHM $\sim 50$ nm centred at 1,550 nm) was used to select the signal and filter out the unwanted background signal. The SSPD was connected to a correlation card (PicoHarp 300) controller, that is, a time-to-digital converter. This controller measures the time between the excitation pulse (the start signal arriving from the trigger signal of the pulse generator) and the arrival of a luminescence photon at the SSPD (stop signal). A histogram of these arrival times is then constructed corresponding to the time-dependent output intensity of the electrically pumped nanopillar LED.

**Data availability.** The data that support the findings of this study are available from the corresponding authors on request.

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

## Acknowledgements

V.D.-C. would like to acknowledge E.J. Geluk, T. de Vries, B. Smalbrugge and H.P.M.M. Ambrosius from NanoLab@TU/e for useful discussions on clean room processing, as well as T. den Dekker from Philips for providing the dicing of the sample. This work was supported by the EU FP7 project NAVOLCHI (288869), the ERC project NO LIMITS, and NanoNextNL, a micro- and nano-technology program of the Dutch Ministry of Economic Affairs and Agriculture and Innovation and 130 partners. B.R. acknowledges the financial support of the Marie Skłodowska-Curie fellowship NANOLASER (2014-IF-659012).

## Author contributions

M.K.S., V.D.-C and A.F. conceived the device structure. D.H. and V.D.-C. performed the device design. P.J.v.V. did the epitaxial growth of the samples. V.D.-C. proposed the fabrication process flow and fabricated the devices. A.H.-R. contributed to process development. F.P., B.R., S.B. and V.D.-C. carried out the characterization. B.R. performed the numerical fitting of the results. V.D.-C., B.R., M.K.S., A.F. and D.H. contributed to the interpretation of the results. V.D.-C and B.R. prepared the manuscript. D.H., A.F. and M.K.S. co-supervised the project.

## Additional information

**Competing financial interests:** The authors declare no competing financial interests.

**Publisher's note**: 

