## [Peer Review File · Nature Communications]

Reviewers' comments:

Reviewer #1 (Remarks to the Author):

In this manuscript, the authors demonstrate a nanoscale light-emitting diode (LED) consisting of a semiconductor nanopillar in a metallic cavity. Light emission from the LED is coupled into an InP waveguide, which is bonded to a silicon substrate. The devices exhibit on-chip external quantum efficiencies (EQE) of 10^{-4} at room temperature and 10^{-2} at 10 K. The recombination lifetimes of the LEDs were measured to be in the sub-ns range using time-resolved photoluminescence (TRPL) and the capabilities of the LEDs for modulation at GHz frequencies were demonstrated.

The device platform developed is novel and would be of interest to the photonics community. However, I need additional information to assess the impact of this work and its suitability for publication in Nature Communications. I am mainly interested on whether this type of devices can be Purcell enhanced to operate at high speeds with high efficiency and whether the energy consumption can be reduced to make them practical devices.

The main limitation of the LEDs demonstrated is that these devices are dominated by a non-radiative recombination mechanism. This mechanism enables GHz speed modulation but decreases the quantum efficiency. Hence, any efforts to improve the efficiency of these LEDs by suppressing the non-radiative recombination mechanism (e.g. using surface passivation techniques) would degrade the modulation speed. This trade-off can be avoided if the LEDs operate in the Purcell-enhanced regime that allows both high modulation speeds and high efficiencies. The current devices have calculated Purcell factors close to unity (considering emitter broadening) and, therefore, cannot operate in the Purcell enhanced regime. The Purcell factor can be increased by improving the emitter-cavity spatial overlap as suggested by the authors. What are the strategies to optimize the emitter-cavity spatial overlap? What is the largest Purcell factor that can be achieved with an optimized emitter-cavity spatial overlap? And, what are the expected efficiencies and modulation speeds with this Purcell factor? Another strategy to improve the Purcell factor is to increase the Q/V of the cavity. What is the mode volume of the current cavities? Can it be reduced to increase the Purcell factor?

The authors measured a low-temperature, on-chip EQE of 10^{-2} - two orders of magnitude higher than at room temperature. What is the measured recombination lifetime at low temperatures? This information can be included in the manuscript for completeness and to highlight the efficiency-speed tradeoff in these devices. Since the surface recombination is completely suppressed at low temperatures (based on the results from the rate equation analysis), one would expect the recombination lifetime to be close to that in bulk (2.7 ns) and, therefore, a 3-dB modulation bandwidth limited to 60 MHz.

The main application of these LEDs is as light sources for optical interconnects. This application has stringent requirements for the energy consumption per transmitted bit? What is the energy consumption per transmitted bit? How does the energy per bit vary as a function of injected current? The energy per bit can be estimated as $E_{\text{bit}} = IV / (1.3f_{3\text{dB}})$ (See Ref. 7: Takeda et al., Nat. Photonics 7, 569, 2013). Using this relation and the I-V data presented in Fig. 3(a), the E_{bit} at a bias current of 100 μA (which yields an output power ~ 1.5 nW with an EQE close to 10^{-4}) is ~ 550 fJ/bit. This is much higher than the target value of 10 fJ/bit for on-chip optical interconnects. The LED can certainly be operated at very low powers to reduce the energy consumption but the output power and EQE would be much lower. The energy consumption can be reduced by reducing the p-contact resistance that results in a high voltage drop. What is the expected improvement by doing so? Are there any other strategies to reduce the power consumption in these devices? Is it possible to reduce the active volume?

Photonic crystal lasers that are waveguide coupled have been demonstrated with low energy consumption, μW level output powers and 10 Gb/s modulation speeds (See Ref. 7: Takeda et al., Nat. Photonics 7, 569, 2013). These lasers have also been integrated on silicon substrates using an O₂-plasma assisted technique (See Takeda et al. Opt. Express 66, 702, 2015). Besides the reduced footprint, are there any other advantages of these LEDs compared to photonic crystal lasers?

The output power of these LEDs (nWs at room temperature) is compared to that of single-mode photonic crystal LEDs (Ref. 16: Shambat et al., Nat. Commun. 2, 539, 2011). Note, however, that the output power of these photonic crystal LEDs is 10s to 100s pW, and not pW as stated in the introduction and discussion sections of the manuscript.

Reviewer #2 (Remarks to the Author):

The authors report on an experimental work on waveguide-coupled metal-covered light-emitting diodes. They design and fabricate III-V-based nanopillars containing an active layer of InGaAs that is electrically excited and coupled to an InP waveguide.

The laser is usually considered the most efficient device to transmit information in the optical domain. However, several groups suggested that using LEDs may be more interesting than lasers for optical interconnects provided that one can circumvent their main limitations, that is: incoherent and isotropic spontaneous emission and relatively slow direct modulation. These two limitations make difficult both the channeling and the fast modulation of light from an LED.

Khurgin et al. theoretically showed that using a metal-coated LED would yield faster, denser and more energy efficient devices than lasers. Furthermore, the group of Yablonovitch, among other groups, theoretically and experimentally demonstrated that metal-antenna-coupled LEDs can be driven faster than lasers.

The present manuscript falls within this subfield with a specific emphasis on technological fabrication and experimental demonstration. It is therefore not truly original, but the main novelty of this work is that the emission from the fabricated LEDs is both fast and coupled to a waveguide mode. Hence, the spontaneous isotropic emission is effectively channeled to a desired direction and directly-modulated at a fast rate, what is a good step towards demonstration of optical interconnect.

The study seems to be conducted seriously and the manuscript contains a wealth of details in the simulation, fabrication and characterization of the devices. However, I have a few concerns that are listed below

1. Fabrication constraints and device requirements

As this paper is highly oriented towards practical implementation and applications, it should be judged by its performances as compared to electrical interconnects. Optical interconnects, to be competitive with metal interconnects must have a low power consumption, be scalable and driven at a high frequency.

The device indeed show a low power consumption but the global process flow is very complicated, with several different number of e-beam lithography, optical lithography, deposition with various techniques and etching. Although this is very impressive, I assume this will be complicated to transfer to the industry. Could the authors comment on that?

The device is bonded on Silicon but this silicon substrate is not used in the study. So why is that important to bond it on silicon in the present case? Could the authors envision a way for coupling the emitted light directly to an SOI substrate? In that case the use of silicon would be clearly an advantage for future use on CMOS platforms integrating routers, photodetectors, etc...

2. Potential improvement and scalability

The active part of the LED has indeed a low footprint but the most important parameter to consider is the total footprint, including electrodes. In the reported device the total footprint is a few hundred microns, which basically represent the same or a bigger footprint than lasers. The authors should comment on that and explain whether such devices could be made more compact.

3. Reproducibility and yield?

For future applications or industrial transfer, it is important to know if the technological processes and fabrication are reproducible. So it should be mentioned whether the results were obtained on a few 'lucky' samples or if it has been measured on several different devices.

4. Modulation and comparison to other directly modulated light-sources

It is explained that the fast recombination lifetimes are dominated by non-radiative recombination, what is the main reason why the sub-nanosecond regime is reached. It means that there is a trade-off between speed and efficiency. In other words, if the authors manage to reduce the non-radiative recombination (i.e. increase the quantum efficiency of their LEDs), they will reduce the modulation bandwidth. This is a big issue. I assume that the initial idea of using metal-cavity was to increase the emission rate through the Purcell effect. But in the end, the presence of metal-cladding is just a way to avoid light leakage from the emitting pillar. The authors should explain

whether it could be possible to decrease the non-radiative recombination and enhance the radiative rate at the same time using a proper passivation layer and a good metal cladding design. This is important to understand if the presented device could be further improved or has already reached its limits.

There exist other efficient ways to directly modulate an integrated light-source, for example using 2D materials [Nat. Physics, 11, 281, (2015)], phase-change materials [Nat. Commun. 6, 8636 (2015)], or Auger quenching [Appl. Phys. Lett. 92, 091103 (2008)]. The authors should compare their results to these other techniques and discuss the respective performances of all these techniques.

In conclusion, this is an interesting paper that is technologically impressive but I believe the different point I raised should be addressed before considering this paper for publication in Nature Communications.

Reviewers' comments:

Reviewer #1 (Remarks to the Author):

In this manuscript, the authors demonstrate a nanoscale light-emitting diode (LED) consisting of a semiconductor nanopillar in a metallic cavity. Light emission from the LED is coupled into an InP waveguide, which is bonded to a silicon substrate. The devices exhibit on-chip external quantum efficiencies (EQE) of 10^{-4} at room temperature and 10^{-2} at 10 K. The recombination lifetimes of the LEDs were measured to be in the sub-ns range using time-resolved photoluminescence (TRPL) and the capabilities of the LEDs for modulation at GHz frequencies were demonstrated.

The device platform developed is novel and would be of interest to the photonics community. However, I need additional information to assess the impact of this work and its suitability for publication in Nature Communications. I am mainly interested on whether this type of devices can be Purcell enhanced to operate at high speeds with high efficiency and whether the energy consumption can be reduced to make them practical devices.

The main limitation of the LEDs demonstrated is that these devices are dominated by a non-radiative recombination mechanism. This mechanism enables GHz speed modulation but decreases the quantum efficiency. Hence, any efforts to improve the efficiency of these LEDs by suppressing the non-radiative recombination mechanism (e.g. using surface passivation techniques) would degrade the modulation speed. This trade-off can be avoided if the LEDs operate in the Purcell-enhanced regime that allows both high modulation speeds and high efficiencies. The current devices have calculated Purcell factors close to unity (considering emitter broadening) and, therefore, cannot operate in the Purcell enhanced regime. The Purcell factor can be increased by improving the emitter-cavity spatial overlap as suggested by the authors. What are the strategies to optimize the emitter-cavity spatial overlap? What is the largest Purcell factor that can be achieved with an optimized emitter-cavity spatial overlap? And, what are the expected efficiencies and modulation speeds with this Purcell factor? Another strategy to improve the Purcell factor is to increase the Q/V of the cavity. What is the mode volume of the current cavities? Can it be reduced to increase the Purcell factor?

A1. We are grateful to the referee for her/his detailed report and for the amount of work dedicated to reviewing our paper and assess its impact and suitability for publication in Nature Communications. In the revised version, we clarified the points raised by the Reviewer, specifically the strategies to mitigate the limitations of the non-radiative recombination mechanism and improve both the efficiency and the speed of our nanoLEDs.

As mentioned by the Reviewer, and as discussed in the original manuscript, the reported nanocavity LEDs do not operate in a Purcell-enhanced regime. Nevertheless, strategies in both the design and fabrication that do not require substantial modifications of the currently proposed metal-cavity nanopillar devices are within reach to achieve both high modulation speeds and higher efficiencies. As suggested by the Reviewer, these methods are now discussed in detailed in our revised manuscript and Supplementary Information, and can be summarized as follows:

- a) Purcell enhancement of the spontaneous emission: both efficiency and modulation speed of the nanoLEDs can be further improved by either optimizing the emitter-cavity spatial overlap or reducing the mode volume of the cavity to achieve Purcell enhancement of our nanoLEDs. Regarding the optimization of the emitter-cavity spatial overlap, there are a number of improvements that can be achieved to improve both lateral and vertical confinement without substantial changes of the reported nanopillars. This discussion can be found in the revised manuscript in the Discussion section and include the following 4 key aspects:

a1) varying the bottom InP post of the nanopillar for vertical confinement - the bottom InP post height has a strong influence in the emitter-cavity spatial overlap due to the presence of the BCB substrate, which has a lower refractive index than the semiconductor, and therefore contributes to the vertical confinement. Therefore, by decreasing the post height, the mode extends less in the InP post and the confinement increases. Although the Q-factor may be reduced, this solution provides the better compromise between better confinement while maintaining a similar waveguide coupling efficiency reported in our manuscript;

a2) Decreasing the insulation thickness for lateral confinement - by decreasing the insulator thickness, losses may become higher via greater interaction with the metal cladding, however a higher overlap of the mode with the InGaAs core can be achieved.

a3) Sidewall angle of the nanopillars - The mode is not as well confined in the InGaAs layer and has a lower Q than for the ideal case. We estimate an angle deviation of about 2° in the real devices when compared with the ideal case of nanopillars with vertical sidewalls.

Besides a reduction of the expected quality factor of the cavity to values <100 , FDTD

simulations show a strong reduction of the Purcell factor from a value close to 12 in an ideal vertical nanopillar down to around 2 assuming an angle deviation of 2° (not accounting with homogenous broadening effect). This highlights the need for control over the pillar sidewall profile during manufacture. A higher verticality (sidewall angle below 1°) has been reported in literature when using $\text{Cl}_2:\text{H}_2:\text{Ar}$ inductively coupled plasma reactive ion etching (RIE), we however used a $\text{CH}_4:\text{H}_2$ based RIE (Ref. [R1]). This discussion, together with a1) and a2) is discussed in the revised manuscript in page 15, line 305.

a4) Decreasing the mode volume – the calculated mode volumes of our nanocavities (revised Supplementary, page 23) indicate the clear possibility of further downscaling our devices to sizes comparable with the smallest mode volumes reported in the literature of comparable metal-dielectric nanocavity devices [9]. While testing experimentally the effects of **a1)-a4)** in the performance of our nanoLEDs cannot be realized immediately without starting the fabrication of new devices, in the revised Supplementary Information, section 7), we present a detailed theoretical treatment that allow us to estimate the effects of the emitter-cavity spatial overlap and the reduction of the mode volume of the cavity in the nanoLEDs performance. Our rate equations model, substantially updated and revised in the revision process, provides a physical description of the rate of photon emission for a homogeneously broadened emitter in a resonant cavity and is derived directly from Fermi's golden rule. Supplementary Fig. 9. summarizes the results of our findings showing that a nanoLED with improved surface passivation (see discussion below) together with device downscaling can potentially operate at Gb/s rates with on-chip optical power levels above 1 μW and energy consumptions below 20 fJ/bit. The results clear demonstrate that the speed-efficiency trade-off can be strongly mitigated. The main results of our analysis are summarized in the main manuscript in page 17, line 328). We note that a single Purcell factor cannot be defined in the case of a bulk active material which is spatially and spectrally distributed - this is why we avoid mentioning Purcell factors in the context of the full model.

- b) NanoLED high-speed switch-off: other methods to improve the speed that go beyond the limitation of the radiative recombination rates and which do not require a Purcell enhanced nanoLED include using reverse-biasing the nanoLED during the turn-off cycle to shorten the minority carrier storage time. This effect has been tested experimentally in our nanoLEDs and the results are included in the revised Supplementary material, section 8.2. Operating the nanoLED at room temperature, we demonstrate a substantial decrease of the switch-off time to ~ 120 ps (Supplementary Fig. 11), that is, much faster than the fastest switch-off time achieved by the nanoLED when its operation is dominated by nonradiative

recombination (~ 290 ps). One of the advantages of this method is that we can achieve a high-speed turn-off of the nanoLED without requiring further changes in the reported nanoLEDs. A disadvantage of this method is that the turn-on of the diode is still limited by the radiative/nonradiative recombination rates. However, techniques to improve the turn-on cycle of the diode can be also implemented, which include current peaking during the turn-on phase or using a larger driving voltage corresponding to a larger quasi-static current (see Ref. [R2]). The above discussion can be found in page 17, line 334 of the revised manuscript.

- c) Surface passivation: improving the efficiency of our nanoLEDs is already within reach by suppressing the non-radiative recombination mechanism, specifically using a surface passivation technique that was recently developed in our group, see Ref. [R3]. Using this improved method based on wet-chemical ammonium sulfide treatment and SiO_x encapsulation, we report a reduction of the surface velocity of two orders of magnitude (~ 500 cm/s) for nanopillars with similar dimensions and using identical epilayer material as the nanoLEDs reported in our manuscript. A theoretical analysis of the performance of the nanoLEDs using a full rate equations model shows that such reduction in the surface velocity results in up to 100-fold increase of the efficiency of our nanoLEDs for low and moderate bias injection current conditions (see Supplementary Information, Supplementary Fig. 9a). Therefore, this efficiency improvement in combination with the modulation speed methods presented above in a) and b) can strongly mitigate the current trade-off between speed and efficiency limited by the non-radiative recombination mechanism. The discussion regarding the improved surface passivation can be found in the revised manuscript in page 15, line 295)

[R1] Parker, J. S., Norberg, E. J., Guzzon, R. S., Nicholes, S. C., & Coldren, L. A. (2011). High verticality InP/InGaAsP etching in Cl₂/H₂/Ar inductively coupled plasma for photonic integrated circuits. *Journal of Vacuum Science & Technology B: Microelectronics and Nanometer Structures*, 29(1), 011016.

[R2] Halbritter, H., Jager, C., Weber, R., Schwind, M. & Mollmer, F. High-Speed LED Driver for ns-Pulse Switching of High-Current LEDs. *IEEE Photonics Technology Letters* 26, 1871–1873 (2014).

[R3] Higuera Rodriguez, A. *et al.* Ultra-low surface recombination for deeply etched III-V semiconductor nano-cavity lasers. in *Advanced Photonics 2016 (IPR, NOMA, Sensors, Networks, SPPCom, SOF)* ITu2A.2 (Optical Society of America, 2016). doi:10.1364/IPRSN.2016.ITu2A.2

The authors measured a low-temperature, on-chip EQE of 10^{-2} - two orders of magnitude higher than at room temperature. What is the measured recombination lifetime at low temperatures? This information can be included in the manuscript for completeness and to highlight the efficiency-speed tradeoff in these devices. Since the surface recombination is completely suppressed at low temperatures (based on the results from the rate equation analysis), one would expect the recombination lifetime to be close to than in bulk (2.7 ns) and, therefore, a 3-dB modulation bandwidth limited to 60 MHz.

A2. As suggested by the Reviewer, we have included in the revised manuscript the experimental results with the measurements of the recombination lifetimes at low temperatures (9.5 K). As shown in the results included in the revised supplementary material [see Supplementary Fig 10(a)], at low-temperature the nanoLED modulation speed is ~ 1.1 ns, that is, the non-radiative losses can be considered negligible since this lifetime compares with the typical nanosecond-range radiative lifetime of the intrinsic InGaAs active material, as confirmed by the simulation results employing the rate equations model, Supplementary Figure 10(b). The discussion of these results can also be found in the revised manuscript in page 13, line 248.

The main application of these LEDs is as light sources for optical interconnects. This application has stringent requirements for the energy consumption per transmitted bit? What is the energy consumption per transmitted bit? How does the energy per bit vary as a function of injected current? The energy per bit can be estimated as $E_{\text{bit}} = IV / (1.3f_{\text{3dB}})$ (See Ref. 7: Takeda et al., Nat. Photonics 7, 569, 2013). Using this relation and the I-V data presented in Fig. 3(a), the E_{bit} at a bias current of 100 μA (which yields an output power ~ 1.5 nW with an EQE close to 10^{-4}) is ~ 550 fJ/bit. This is much higher than the target value of 10 fJ/bit for on-chip optical interconnects. The LED can certainly be operated at very low powers to reduce the energy consumption but the output power and EQE would be much lower. The energy consumption can be reduced by reducing the p-contact resistance that results in a high voltage drop. What is the expected improvement by doing so? Are there any other strategies to reduce the power consumption in these devices? Is it possible to reduce the active volume?

A3. We agree, this is an important point. In Supplementary Fig. 8 we provide additional information regarding the operating energy versus injected current extracted from our measurements. The results show that for the measurements at room temperature of the device reported in Fig. 4, in the current range of 1 μA to 30 μA our nanoLED displays an energy consumption ranging from ~ 1 fJ/bit

to 100 fJ/bit. In our experiments the nanoLEDs were operated at moderate and low bias pumping conditions (<40 μA) mainly to avoid devices' thermal heating and possible failure. The estimated energy consumption if the nanoLED is operated at around its maximum output optical power at room temperature (bias current of $\sim 100 \mu\text{A}$) is 540 fJ/bit, a similar value indicated by the Reviewer. The discussion of these results can be found in the revised manuscript in page 14, line 272. Although we agree with the Reviewer that operating the currently reported nanoLEDs at their maximum optical power yields an energy consumption much higher than the 10 fJ/bit value required in optical interconnects, as discussed in detail in the Comment A1., a number of strategies (namely surface passivation and reducing the mode volume) are within reach to further improve the nanoLEDs' efficiency, allowing μW -level output power at much lower current bias levels without compromising the operation speed (see Supplementary Figure 9), and thereby reducing the energy/bit to the few tens of fJ level.

Additionally, the energy consumption can be reduced by improving the ohmic contacts as mentioned by the Reviewer. In this respect, our group has recently reported ultra-low ohmic contacts based on germanium-silver, as used in our nanoLEDs, on heavily doped n-type InGaAsP layers (Ref. [R4]), decreasing the contact resistance about one order of magnitude ($< 1 \times 10^{-7} \Omega\text{cm}^2$) with respect to our devices. On the p-side, the series resistance can be reduced by increasing the thickness of the p-doped InGaAsP layer (only 100 nm thick in our devices). A brief discussion regarding the improvement of the ohmic contacts can be found in page 15, line 300 of the revised manuscript.

[R4] Shen, L., Van Veldhoven, P. J., Jiao, Y., Dolores-Calzadilla, V., Van Der Tol, J. J. G. M., Roelkens, G., & Smit, M. K. (2016). Ohmic contacts with ultra-low optical loss on heavily doped n-type InGaAs and InGaAsP for InP-based photonic membranes. *IEEE Photonics Journal*, 8(1).

Photonic crystal lasers that are waveguide coupled have been demonstrated with low energy consumption, μW level output powers and 10 Gb/s modulation speeds (See Ref. 7: Takeda et al., Nat. Photonics 7, 569, 2013). These lasers have also been integrated on silicon substrates using an O₂-plasma assisted technique (See Takeda et al. Opt. Express 66, 702, 2015). Besides the reduced footprint, are there any other advantages of these LEDs compared to photonic crystal lasers?

A4. Despite the fact that promising photonic crystals on Si have been recently demonstrated, metal nanocavities can still offer a few advantages that justify a great interest and further investigations on this approach. Although we already discuss in the original manuscript some of the advantages of

nanoscale LEDs compared with lasers for on-chip communication systems (see Introduction section, pages 3 and 4), in the revised manuscript we provide additional information of the advantages of metal-cavity nanoLEDs when compared with photonic crystal lasers (included in page 4, line 60 and page 15, line 284) and also included the Reference [R5] suggested by the Reviewer in the references list:

[R5] Takeda, K. et al. Heterogeneously integrated photonic-crystal lasers on silicon for on/off chip optical interconnects. *Opt. Express* 23, 702–708 (2015).

The key advantages can be summarized as follows. Firstly, as mentioned by the Reviewer, the footprint can be significantly smaller. This can be achieved by either operating with a dielectric mode as reported here or in the plasmonic regime where such kind of structures do not present a mode cut-off with respect to the cavity size.

Secondly, typical light sources bonded on Si-substrates require an intermediate SiO₂ or BCB polymer layer. This imposes a serious limitation in the heat dissipation through the substrate, since both SiO₂ and BCB have a poor thermal conductivity. Embedding the light source in a metal layer helps to spread and sink the heat in an efficient manner. Recent studies have shown the advantages of this approach for heat dissipation in a metal nanocavity with AlSiO₂ dielectric cladding (Ref. [R6]).

Finally, a LED source also allows for a modulation speed beyond the 3dB operation point without a substantial decrease in modulation depth as compared to lasers whose response deteriorates quickly beyond this point. Additionally, unlike a laser which must be pumped hard to reach large bandwidths, large extinction ratios can be maintained at high speeds at low bias current levels which can be attractive for large-signal digital modulation applications. Furthermore, the current levels can be potentially greatly reduced from lasers (including “thresholdless” lasers), which must ‘waste’ additional power for stimulated emission to overcome spontaneous emission.

[R6] Gu, Q., Shane, J., Vallini, F., Wingad, B., Smalley, J. S. T., Frateschi, N. C., & Fainman, Y. (2014). Electrically pumped etallo-dielectric pedestal nanolasers with amorphous Al₂O₃ shield. *IEEE Journal of Quantum Electronics*, 50(7), 499–509.

The output power of these LEDs (nWs at room temperature) is compared to that of single-mode photonic crystal LEDs (Ref. 16: Shambat et al., *Nat. Commun.* 2, 539, 2011). Note, however, that

the output power of these photonic crystal LEDs is 10s to 100s pW, and not pW as stated in the introduction and discussion sections of the manuscript.

A5. We have corrected in the revised manuscript [Introduction (page 4, line 67) and Discussion (page 14, line 283) sections] the optical output power values of the single mode photonic crystal LEDs.

Reviewers' comments:

Reviewer #2 (Remarks to the Author):

The authors report on an experimental work on waveguide-coupled metal-covered light-emitting diodes. They design and fabricate III-V-based nanopillars containing an active layer of InGaAs that is electrically excited and coupled to an InP waveguide.

The laser is usually considered the most efficient device to transmit information in the optical domain. However, several groups suggested that using LEDs may be more interesting than lasers for optical interconnects provided that one can circumvent their main limitations, that is: incoherent and isotropic spontaneous emission and relatively slow direct modulation. These two limitations make difficult both the channeling and the fast modulation of light from an LED. Khurgin et al. theoretically showed that using a metal-coated LED would yield faster, denser and more energy efficient devices than lasers. Furthermore, the group of Yablonovitch, among other groups, theoretically and experimentally demonstrated that metal-antenna-coupled LEDs can be driven faster than lasers.

The present manuscript falls within this subfield with a specific emphasis on technological fabrication and experimental demonstration. It is therefore not truly original, but the main novelty of this work is that the emission from the fabricated LEDs is both fast and coupled to a waveguide mode. Hence, the spontaneous isotropic emission is effectively channeled to a desired direction and directly-modulated at a fast rate, what is a good step towards demonstration of optical interconnect.

The study seems to be conducted seriously and the manuscript contains a wealth of details in the simulation, fabrication and characterization of the devices. However, I have a few concerns that are listed below

1. Fabrication constraints and device requirements

As this paper is highly oriented towards practical implementation and applications, it should be judged by its performances as compared to electrical interconnects. Optical interconnects, to be competitive with metal interconnects must have a low power consumption, be scalable and driven at a high frequency.

The device indeed show a low power consumption but the global process flow is very complicated, with several different number of e-beam lithography, optical lithography, deposition with various techniques and etching. Although this is very impressive, I assume this will be complicated to transfer to the industry. Could the authors comment on that?

B1. We agree with the reviewer that the device fabrication is a challenge by itself. Regarding the fabrication process, an industrial environment would actually offer more stable and reproducible processes. We note that the smallest feature size in our nanoLED is 300 nm, much larger than typical values of present electronic ICs, and comparable to the feature sizes used in silicon photonics. In our case, e-beam lithography was used in order to circumvent the limited resolution of optical lithography systems typically available in academia. In fact, our nano-LED devices could be developed using other lithographic methods widely employed in photonics, namely deep UV lithography, Ref. [S1], widely used in SOI structures which already provide resolutions compatible with the smallest features of our devices. This information has been highlighted and the reference [S1] has been included in the revised Supplementary material, page 3, line 51.

Furthermore, the number of process steps, certainly substantial for an academic fabrication process, is not particularly large when compared to industrial processes for photonic or electronic ICs. We also note that the device yield in our process was very high (~100% for all the devices tested), indicating that it is by no way an irreproducible process. We therefore believe that industrialization is possible, especially taking into account the expected evolution of electronic and photonic fabs in the coming years.

[S1] W. Bogaerts, *et. al*, Fabrication of photonic crystals in silicon-on-insulator using 248-nm deep UV lithography, IEEE Journal of Selected Topics in Quantum Electronics 8(4):928 – 934, August 2002.

The device is bonded on Silicon but this silicon substrate is not used in the study. So why is that important to bond it on silicon in the present case? Could the authors envision a way for coupling the emitted light directly to an SOI substrate? In that case the use of silicon would be clearly an advantage for future use on CMOS platforms integrating routers, photodetectors, etc...

B2. We appreciate the opportunity to clarify some aspects of the photonic platform. As highlighted in the revised manuscript (page 5, line 75) and Supplementary (page 4, line 56) we envision the use of nanoscale sources for IMOS (InP-Membranes On Silicon). In this photonic platform, all the optical functionality is performed in a “photonics plane” consisting of a III-V layer stack, whereas the Si-

substrate hosts the “electronics plane” where electronic components will be placed in the near future. We are indeed currently working to include embedded electronics on our IMOS chips.

In the device presented in our manuscript, the Si-substrate is relevant because of the following key reasons: (i) it shows that our nano-LEDs are fully compatible with our IMOS platform, and therefore can be integrated with other compact passive and active components we have demonstrated in such a platform. (ii) Components such as light sources and waveguides show the best compatibility and performance using InP semiconductor materials. (iii) The use of the bonding layer is also important because it allows an InP-waveguide with high index (i.e. with compact cross section) which facilitates the coupling with the nanocavity mode with high confinement, one of the key aspects of our approach.

From the technology point of view, this device can also be integrated with SOI. In that case, the waveguide (and other passive components) can be fabricated at first in a SOI wafer and then a III-V layer stack can be bonded on top with a thin BCB layer (50 nm bonding has been previously demonstrated, Ref. [R7]). Finally, the nanopillar LED would be fabricated on the III-V layer stack. This aspect is now highlighted in the revised manuscript, page 5, line 93. An alternative approach could consist in using a thick BCB bonding layer and let the LED emit vertically similar to a VCSEL, then the light could be coupled to an underlying SOI chip by means of grating couplers.

[R7] Roelkens, G., Liu, L., Liang, D., Jones, R., Fang, A., Koch, B., & Bowers, J. (2010). III-V/silicon photonics for on-chip and intra-chip optical interconnects. *Laser and Photonics Reviews*, 4(6), 751–779.

2. Potential improvement and scalability

The active part of the LED has indeed a low footprint but the most important parameter to consider is the total footprint, including electrodes. In the reported device the total footprint is a few hundred microns, which basically represent the same or a bigger footprint than lasers. The authors should comment on that and explain whether such devices could be made more compact.

B3. The footprint of our devices is indeed large (280 μm x 170 μm), however this is only for practical reasons related to our characterisation set-ups. Our electrodes were designed to have a separation distance of 100 μm , since this allows an easier characterization with standard G-S-G electrical probes that are available in our electro-optical characterization setup. In a practical application, the electrodes can be as small as the technology for electrical interconnecting to the CMOS chip allows.

We also envision the possibility to perform metal bonding between the electrodes in a Si-substrate (and electronics in the future) and the metal-cavity LEDs in the III-V stack (flip-chip); in this approach, the electrodes could be as small as the metal nanocavity if allowed by the accuracy of the bonding machine. This discussion can be found in page 9, line 172 of the revised manuscript.

3. Reproducibility and yield?

For future applications or industrial transfer, it is important to know if the technological processes and fabrication are reproducible. So it should be mentioned whether the results were obtained on a few 'lucky' samples or if it has been measured on several different devices.

B4. As the cavity resonance frequency is very sensitive to nanopillar size, indeed we fabricated a series of devices with different symmetric cross sections ranging between 300 nm x 300 nm and 400 nm x 400 nm. Although the experimental results of the static and dynamic characterization presented in the main manuscript were obtained on nanopillars with side length of 350 nm and 340 nm (they were chosen since they operate at the target wavelengths ~ 1550 nm), as can be checked by the additional L-I characteristics included in the revised supplementary material (Supplementary Fig. 7) the remaining tested nanocavities (tens of devices were tested) with identical horizontal cross sections displayed similar performances in terms of optical output and maximum external quantum efficiencies indicating a high yield. The discussion of these results can also be found in the revised manuscript in page 10, line 197. Unfortunately, at this early research stage on the development of our waveguide-coupled metal nanocavity LED devices we do not have a clear indication regarding their reproducibility. However, we cannot identify any critical steps or incompatibility in our fabrication process with currently existent photonics/silicon photonics industrial foundries that suggest serious problems with reproducibility.

4. Modulation and comparison to other directly modulated light-sources

It is explained that the fast recombination lifetimes are dominated by non-radiative recombination, what is the main reason why the sub-nanosecond regime is reached. It means that there is a trade-off between speed and efficiency. In other words, if the authors manage to reduce the non-radiative recombination (i.e. increase the quantum efficiency of their LEDs), they will reduce the modulation bandwidth. This is a big issue. I assume that the initial idea of using metal-cavity was to increase the emission rate through the Purcell effect. But in the end, the presence of

metal-cladding is just a way to avoid light leakage from the emitting pillar. The authors should explain whether it could be possible to decrease the non-radiative recombination and enhance the radiative rate at the same time using a proper passivation layer and a good metal cladding design. This is important to understand if the presented device could be further improved or has already reached its limits.

B5. Improving the efficiency is indeed already within reach by suppressing the non-radiative recombination mechanism, specifically using a surface passivation technique that was recently developed in our group, see Ref. [R3]. Using this improved method based on wet-chemical ammonium sulfide treatment and SiO_x encapsulation, we report a reduction of the surface velocity of two orders of magnitude (~500 cm/s) for nanopillars with similar dimensions and using identical epitaxial layer stack as the nanoLEDs reported in our manuscript. A theoretical analysis of the performance of the nanoLEDs using a full rate equations model shows that such reduction in the surface velocity results in up to 100-fold increase of the efficiency of our nanoLEDs for low and moderate bias injection current conditions. From these calculations we conclude that at injection levels in the μA range, Auger nonradiative recombination enables reaching a GHz-range bandwidth (see Supplementary Information, Fig. 9a), providing a good efficiency-bandwidth trade-off. In the longer term, further downscaling of the nano-LED and fabrication improvements can allow entering the Purcell-enhanced regime, with substantial benefits in efficiency and bandwidth, as discussed in the detail in the reply to Reviewer #A1 and reported in revised manuscript in Discussion section and in more detail in Section 7.4 of the revised Supplementary material.

[R3] Higuera Rodriguez, A. et al. Ultra-low surface recombination for deeply etched III-V semiconductor nano-cavity lasers. in *Advanced Photonics 2016 (IPR, NOMA, Sensors, Networks, SPPCom, SOF) ITu2A.2* (Optical Society of America, 2016). doi:10.1364/IPRSN.2016.ITu2A.2

There exist other efficient ways to directly modulate an integrated light-source, for example using 2D materials [Nat. Physics, 11, 281, (2015)], phase-change materials [Nat. Commun. 6, 8636 (2015)], or Auger quenching [Appl. Phys. Lett. 92, 091103 (2008)]. The authors should compare their results to these other techniques and discuss the respective performances of all these techniques.

B6. As suggested by the Reviewer, our modulation technique and performance has been compared with the modulation techniques mentioned above in the revised manuscript in page 16, line 313. The references suggested by the Reviewer have been also added to the revised manuscript:

[R8] Tielrooij, K. J. et al. Electrical control of optical emitter relaxation pathways enabled by graphene. Nat Phys 11, 281–287 (2015).

[R9] Cueff, S. et al. Dynamic control of light emission faster than the lifetime limit using VO₂ phase-change. Nat Commun 6, (2015).

[R10] Carreras, J., Arbiol, J., Garrido, B., Bonafos, C. & Montserrat, J. Direct modulation of electroluminescence from silicon nanocrystals beyond radiative recombination rates. Appl. Phys. Lett. 92, (2008).

In conclusion, this is an interesting paper that is technologically impressive but I believe the different point I raised should be addressed before considering this paper for publication in Nature Communications.

- **Additional minor changes in the revised manuscript and supplementary information:**

1. *The previous rate equations model was removed from the methods section. The model and theoretical description of the nano-LED using rate equations was updated to take into account the Purcell enhancement of the spontaneous emission. The full details of the model are now included in the revised Supplementary information, section 7. The corresponding model fittings in Figures 3b and 4a were updated using the revised and updated rate equations model.*

2. *The equation presented in Supplementary material, page 13, line 171, used to estimate the surface velocity was corrected from the previous version to take into account the case of a nanopillar. The corresponding estimated value of the surface passivation was also corrected (see Supplementary, page 13, line 173).*

3. *Considering our recent developments in the surface passivation experiments, the sentence “An improved passivation might be achieved by replacing the SiO₂ cladding by Si₃N₄ and optimizing the thermal annealing conditions for recovering the surface damage induced by the plasma enhanced chemical vapor deposition (PECVD) of the dielectric cladding” was removed from the revised manuscript.*

Reviewers' comments:

Reviewer #1 (Remarks to the Author):

One of the major issues in the demonstrated LEDs is the trade-off between efficiency and modulation speed. In the revised manuscript, the authors discuss how these devices can be improved to operate in the Purcell-enhanced regime--with both high efficiency and at high speeds--by optimizing the emitter-cavity overlap and by reducing the mode volume. The overlap can be optimized by reducing the height of the bottom InP post, reducing the thickness of the dielectric surrounding the nanopillar or improving the pillar etching step to achieve higher verticality (with sidewall angles below 1°). The mode volume can be reduced by reducing the device dimensions. The authors also discuss how the efficiency of the LEDs can be improved by passivating the surface of the nanopillar and proposed the use of a surface passivation technique developed in their group. This technique, which is based on an ammonium sulfide treatment followed by SiOx encapsulation, reduces the surface recombination velocity by two orders of magnitude. The revised manuscript also includes the demonstration of faster switch-off times by reverse biasing the LEDs and an improved rate equation model.

Even though the demonstrated LEDs do not operate with both high efficiency and at high speed, the authors have proposed strategies to improve the devices such that operation in the Purcell-enhanced regime is possible. If the implementation of these strategies is successful, these LEDs would potentially meet the requirements for on-chip optical interconnects. This novel device platform has some advantages over other platforms proposed for optical interconnects and, therefore, it would be of interest to the photonics community. I recommend this manuscript for publication in Nature Communications if the following points are addressed.

The authors propose the use of reduced dielectric thickness and post heights to improve the emitter-cavity overlap. Can this improvement be quantified? What is the overlap as a function of post height and dielectric thickness? In the analysis presented in supplementary section 7.4, a perfect emitter-cavity overlap is assumed. What is the maximum emitter-cavity overlap possible considering the field spatial distribution shown in Fig. 1?

The authors show the energy consumption per transmitted bit as a function of injected current in Supplementary Fig. 8. The maximum current in the figure is below the current for maximum output power. What is the optical power and external differential efficiency in this current range? It is important to add this information to this figure.

The authors demonstrate faster switch-off times by reverse biasing the LED. The authors proposed that the switch-on time can be reduced by using the current peaking technique described by Halbritter et al. (Ref. R2). Can the authors implement this experimentally?

The analysis presented in Supplementary section 7.4 for devices with 10x lower mode volumes yields energy consumptions below 20 fJ/bit. What are the device dimensions required to achieve an energy per bit of less than 10 fJ/bit (as required by on-chip optical interconnects)? Are these dimensions practically realizable?

The authors calculated a Purcell factor of 2 for pillars with sidewall angles of 2° . What is the Purcell factor for pillars with sidewall angles of 1° ?

Reviewer #2 (Remarks to the Author):

The authors have addressed all my previous concerns and I believe the manuscript is now suitable for publication in Nature Communications.

Reviewers' comments:

Reviewer #1 (Remarks to the Author):

One of the major issues in the demonstrated LEDs is the trade-off between efficiency and modulation speed. In the revised manuscript, the authors discuss how these devices can be improved to operate in the Purcell-enhanced regime--with both high efficiency and at high speeds--by optimizing the emitter-cavity overlap and by reducing the mode volume. The overlap can be optimized by reducing the height of the bottom InP post, reducing the thickness of the dielectric surrounding the nanopillar or improving the pillar etching step to achieve higher verticality (with sidewall angles below 1°). The mode volume can be reduced by reducing the device dimensions. The authors also discuss how the efficiency of the LEDs can be improved by passivating the surface of the nanopillar and proposed the use of a surface passivation technique developed in their group. This technique, which is based on an ammonium sulfide treatment followed by SiO_x encapsulation, reduces the surface recombination velocity by two orders of magnitude. The revised manuscript also includes the demonstration of faster switch-off times by reverse biasing the LEDs and an improved rate equation model.

Even though the demonstrated LEDs do not operate with both high efficiency and at high speed, the authors have proposed strategies to improve the devices such that operation in the Purcell-enhanced regime is possible. If the implementation of these strategies is successful, these LEDs would potentially meet the requirements for on-chip optical interconnects. This novel device platform has some advantages over other platforms proposed for optical interconnects and, therefore, it would be of interest to the photonics community. I recommend this manuscript for publication in Nature Communications if the following points are addressed.

The authors propose the use of reduced dielectric thickness and post heights to improve the emitter-cavity overlap. Can this improvement be quantified? What is the overlap as a function of post height and dielectric thickness? In the analysis presented in supplementary section 7.4, a perfect emitter-cavity overlap is assumed. What is the maximum emitter-cavity overlap possible considering the field spatial distribution shown in Fig. 1?

A1. As described in our rate equations model, the spontaneous emission rate into the cavity mode is reduced by a factor of $V_{a,eff}/V_a$ as a result of the spatial dependence of the electric field in the active region in our metal nanocavity. For the ideal case of a nanocavity with vertical sidewalls, as represented in the mode profile of Fig. 1c, we calculate $V_{a,eff}/V_a \sim 0.48$, and this information has been included in the revised Supplementary information, page 23. As discussed below, we note that this value has been already optimized and it is close to the maximum emitter-cavity overlap possible for this particular design and nanopillar dimensions. However, as mentioned by the Reviewer, further optimization can be achieved by changing the post height or/and the thickness of the dielectric. When the dielectric thickness is reduced from the design value (175 nm) down to 120 nm, we calculate an improvement up to 0.55, although in this particular case the cavity resonant wavelength also suffers a shift from the target wavelength at 1550 nm to ~ 1480 nm. This trade-off can be compensated by adjusting the semiconductor nanopillar width in order to keep the cavity resonance at the target wavelength.

For the case of changing the post height, we would like to clarify that this value should be varied once the optimum value of insulating thickness is found. For the case of the design value (175 nm), since the post height has been already optimized, a reduction of its thickness does not substantially modifies the emitter-cavity overlap. Alternatives to our design that could make substantial improvements in the emitter-cavity overlap include developments of ultrathin III-V active layers in buried heterostructures [R2.1] or nanoscale metallic coaxial devices composed by a metallic rod surrounded by a metal-coated semiconductor ring [R2.2].

As suggested by the Reviewer, the discussion above and the references below have been included in the revised Supplementary information, page 23.

[R2.1] Shinji Matsuo, Takuro Fujii, Koichi Hasebe, Koji Takeda, Tomonari Sato, and Takaaki Kakitsuka, "Directly modulated buried heterostructure DFB laser on SiO₂/Si substrate fabricated by regrowth of InP using bonded active layer," *Opt. Express* 22, 12139-12147 (2014).

[R2.2] M. Khajavikhan, A. Simic, M. Katz, J. H. Lee, B. Slutsky, A. Mizrahi, V. Lomakin and Y. Fainman, "Thresholdless nanoscale coaxial lasers," *Nature* 482, 204–207 (2012).

The authors show the energy consumption per transmitted bit as a function of injected current in Supplementary Fig. 8. The maximum current in the figure is below the current for maximum output power. What is the optical power and external differential efficiency in this current range? It is important to add this information to this figure.

A2. As suggested by the Reviewer, we have included the panel (b) in the revised Supplementary Fig. 8, with the corresponding information regarding the optical power and external quantum efficiency.

The authors demonstrate faster switch-off times by reverse biasing the LED. The authors proposed that the switch-on time can be reduced by using the current peaking technique described by Halbritter et al. (Ref. R2). Can the authors implement this experimentally?

A3. As reported in R2 (and also R2.4), current peaking techniques have been demonstrated in a driving electronics circuitry using either a speed-up capacitance in parallel to the current-limiting resistor [R2] or using a parallel Schottky-capacitance circuit in series with the LED light source [R2.4]. Although our IMOS platform targets the integration of electronic and photonic circuits, we are currently still working to demonstrate such embedded electronics on the Si-substrate. When we succeed in this, it will certainly be possible to integrate electronic drivers as the ones proposed by Halbritter et al. next to photonic circuits and physically close enough not to limit their high-speed performance. This discussion has been included in the revised manuscript (see Supplementary Information, page 28) and Reference [R2.4] was added to the Supplementary Information. Another possibility would be to do wire bonding of the nanoLEDs to a driving circuitry. However, since we did not plan for that, our current sample is unfortunately not suitable for wire bonding due to the configuration and location of the contact pads that would require gold wires $\gg 10$ mm length which will in any case limit substantially the driving speed due to wires' parasitics (for each 1-2 mm an additional parasitic inductance of ~ 1 nH must be considered).

[R2] Halbritter, H., Jager, C., Weber, R., Schwind, M. & Mollmer, F. High-Speed LED Driver for ns-Pulse Switching of High-Current LEDs. IEEE Photonics Technology Letters 26, 1871–1873 (2014).

[R2.4] P. H. Binh, P. Renucci, V. G. Truong and X. Marie, "Schottky-capacitance pulse-shaping circuit for high-speed light emitting diode operation," in Electronics Letters, vol. 48, no. 12, pp. 721-723, June 7 2012.

The analysis presented in Supplementary section 7.4 for devices with 10x lower mode volumes yields energy consumptions below 20 fJ/bit. What are the device dimensions required to achieve an energy per bit of less than 10 fJ/bit (as required by on-chip optical interconnects)? Are these dimensions practically realizable?

A4. Regarding the energy consumption, we would like to highlight that operation <10 fJ/bit is already predicted by the numerical simulations in the conditions presented in Supplementary Fig. 9, if we operate the nano-LED slightly below $1 \mu\text{W}$ output power, that is, in the current range of $1\text{-}4 \mu\text{A}$. In this situation, the corresponding energy consumption is 8 to 10.2 fJ/bit, respectively, and therefore in the range required by on-chip optical interconnects. This information has been included in the revised manuscript (see page 26 of the Supplementary Information). While theoretically there is a clear path towards achieving nanoLEDs operating at the target energy efficiency for interconnects (i.e. <10 fJ/bit) via mode volume reduction, we also recognize that further reducing the dimension of the pillars well below 150 nm to achieve small mode volumes is technologically challenging without considering changes in our current nanoLED design (see, e.g., refs. [R2.1] and [R2.2]). Practical issues include an increased difficulty to obtain straight sidewall angles due to a strong mask erosion in the high aspect ratio hardmask needed to deeply etch pillars with small cross section. Additionally, the nanopillars might be under a large stress if the thickness of the metal cladding is significantly larger than the nanopillar diameter. As suggested by the Reviewer, we have described these technological challenges in the revised manuscript (see page 27 of the Supplementary Information).

The authors calculated a Purcell factor of 2 for pillars with sidewall angles of 2° . What is the Purcell factor for pillars with sidewall angles of 1° ?

A5. While FDTD simulations show a strong reduction of the Purcell factor down to 1.7 when assuming sidewall angle of 2° (not accounting with homogenous broadening effect), a sidewall angle of 1° results in a calculated Purcell factor of 3.4 .

Reviewers' comments:

Reviewer #2 (Remarks to the Author):

The authors have addressed all my previous concerns and I believe the manuscript is now suitable for publication in Nature Communications.

REVIEWERS' COMMENTS:

Reviewer #1 (Remarks to the Author):

The revised manuscript by Dolores-Calzadilla et al. now includes a figure of the output power and external quantum efficiency as a function of injected current as well as additional discussions on the emitter-cavity overlap and the challenges of decreasing the mode volume to improve the performance of the device. The figure shows that the output power of the device is in the tens of pW levels when operated below 10 fJ/bit. These power levels are comparable to those of photonic crystal LEDs (Ref 16: Shambat et al., Nat. Commun. 2, 539, 2011).

The manuscript has been improved compared to the original version and I can now recommend it for publication in Nature Communications.

Finally, I encourage the authors to continue the development of this novel device platform to realize a Purcell-enhanced LED capable of operating at high speed with high efficiency.

Response to Referees Letter

REVIEWERS' COMMENTS:

Reviewer #1 (Remarks to the Author):

The revised manuscript by Dolores-Calzadilla et al. now includes a figure of the output power and external quantum efficiency as a function of injected current as well as additional discussions on the emitter-cavity overlap and the challenges of decreasing the mode volume to improve the performance of the device. The figure shows that the output power of the device is in the tens of pW levels when operated below 10 fJ/bit. These power levels are comparable to those of photonic crystal LEDs (Ref 16: Shambat et al., Nat. Commun. 2, 539, 2011).

The manuscript has been improved compared to the original version and I can now recommend it for publication in Nature Communications.

Finally, I encourage the authors to continue the development of this novel device platform to realize a Purcell-enhanced LED capable of operating at high speed with high efficiency.

Authors' comments: We would like to thank the Reviewer for his efforts and valuable suggestions during the revision.